# *ZNF117* regulates glioblastoma stem cell differentiation towards oligodendroglial lineage

Jun Liu[1,7], Xiaoying Wang[1,2,7], Ann T. Chen[1,3,7], Xingchun Gao[1,7], Benjamin T. Himes[3], Hongyi Zhang[1], Zeming Chen[1], Jianhui Wang[4], Wendy C. Sheu[1,3], Gang Deng[1], Yang Xiao [3], Pan Zou[1], Shenqi Zhang [1], Fuyao Liu[1], Yong Zhu[5], Rong Fan [3], Toral R. Patel [1,6✉], W. Mark Saltzman [3✉] & Jiangbing Zhou [1,3✉]

Glioblastoma (GBM) is a deadly disease without effective treatment. Because glioblastoma stem cells (GSCs) contribute to tumor resistance and recurrence, improved treatment of GBM can be achieved by eliminating GSCs through inducing their differentiation. Prior efforts have been focused on studying GSC differentiation towards the astroglial lineage. However, regulation of GSC differentiation towards the neuronal and oligodendroglial lineages is largely unknown. To identify genes that control GSC differentiation to all three lineages, we performed an image-based genome-wide RNAi screen, in combination with single-cell RNA sequencing, and identified *ZNF117* as a major regulator of GSC differentiation. Using patient-derived GSC cultures, we show that *ZNF117* controls GSC differentiation towards the oligodendroglial lineage via the Notch pathway. We demonstrate that *ZNF117* is a promising target for GSC differentiation therapy through targeted delivery of CRISPR/Cas9 gene-editing nanoparticles. Our study suggests a direction to improve GBM treatment through differentiation of GSCs towards various lineages.

[1] Department of Neurosurgery, Yale University, New Haven, CT 06511, USA. [2] Wuxi School of Medicine, Jiangnan University, Wuxi, Jiangsu Province 214122, China. [3] Department of Biomedical Engineering, Yale University, New Haven, CT 06510, USA. [4] Department of Pathology, Yale University, New Haven, CT 06511, USA. [5] School of Public Health, Yale University, New Haven, CT 06511, USA. [6] Department of Neurosurgery, University of Texas Southwestern Medical Center, Dallas, TX, USA. [7] These authors contributed equally: Jun Liu, Xiaoying Wang, Ann T. Chen, and Xingchun Gao. ✉email: toral.patel@utsouthwestern.edu; mark.saltzman@yale.edu; jiangbing.zhou@yale.edu

Glioblastoma (GBM) is the most common and aggressive form of primary brain cancer. Even with aggressive treatment, the prognosis is extremely poor with an overall 5-year survival rate of less than 4%[1]. The therapeutic failures in this disease can be partially attributed to the lack of effective chemotherapeutics. Temozolomide (TMZ) is currently the only front-line agent available for GBM. Despite its effectiveness on differentiated GBM cells, TMZ cannot efficiently kill glioblastoma stem cells (GSCs), which are cells within GBM tumors that are considered to be the root of GBM development[2–5]. Therefore, improved treatment of GBM requires elimination of GSCs.

Elimination of GSCs can be potentially achieved through two different approaches. The most straightforward one is to develop cytotoxic therapeutics to which GSCs are susceptible, such as dithiazanine iodide, which we recently identified[6]. An alternative approach is to induce GSC differentiation, making them susceptible to existing therapeutic interventions[7,8]. Compared to the use of cytotoxic agents, differentiation treatments typically have less toxicity[9–12]. Differentiation therapy has proven successful in the treatment of other cancers. For example, acute promyelocytic leukemia has become a curable disease through the use of all-trans retinoic acid-based differentiation therapy[13].

In this study, using a combination of image-based genome-wide RNAi screening and single-cell RNA sequencing (scRNA-seq), we identified zinc finger protein 117 (ZNF117), also known as H-PLK, as a regulator of GSC differentiation. We show that ZNF117 expression is downregulated in differentiated GSCs and that downregulation of ZNF117 induces GSC differentiation towards the oligodendroglial lineage, leading to inhibition of tumor development and increased sensitivity of GSCs to TMZ treatment. We demonstrate that ZNF117 can be targeted for GBM treatment in mouse xenograft models through targeted delivery of clustered regularly interspaced short palindromic repeats (CRISPR) machinery via nanoparticles (NPs). To determine the molecular mechanism of ZNF117-mediated GSC differentiation, we used a combination of cDNA array and chromatin immunoprecipitation sequencing (ChIP-seq) and found that ZNF117 regulates the Notch pathway through interaction of JAG2. Taken together, our results suggest that ZNF117 is a major regulator of GSC differentiation and can be targeted for GBM differentiation therapy to improve the clinical response to current treatment strategies.

## Results

### Genome-wide RNAi screen for identification of GSC differentiation regulators.
To identify genes that regulate GSC differentiation and viability, we performed a genome-wide RNAi screen using the Dharmacon siGenome library. This library consists of siRNAs targeting 18,119 genes, with four siRNAs specific to each gene (Supplementary Data 1). The primary screen was performed in triplicate using GS5 GSCs, which have been well-characterized by our group and others[6,14,15], through a reverse transfection procedure (Fig. 1a). Five days post-transfection, the cells were fixed and stained with Hoechst 33342, which identifies all cells, and an anti-Nestin antibody, which identifies Nestin+ GSCs. By using a NES (Nestin) siRNA as a positive control and a RISC-free siRNA as a negative control, we were able to confirm up to 93% knockdown of the targeted genes (Fig. 1b, Supplementary Fig. 1). Results were then sorted based on a statistically significant reduction of the Nestin+ cell population, as well as the total cell population. The top 100 genes, which demonstrated the greatest effect in reducing the number of total cells and/or the number of Nestin+ cells, were selected (Supplementary Table 1). These genes underwent a secondary validation screen, where transfections were repeated using a single siRNA per well rather than pooled siRNAs targeting a specific

gene. Transfection and staining were performed in an identical manner to the primary screen in triplicate but with four siRNAs for each gene. Control groups were treated with RISC-free siRNA. In the end, all genes were ranked based on their ability to regulate the percentage of total cells (cytotoxicity group) or the percentage of Nestin+ cells (differentiation group) (Supplementary Table 1). As expected, most genes in the cytotoxicity group were involved in cell cycle regulation, such as CCNB1 and WEE1, or are components in the ubiquitin–proteasome system (UPS), such as UBC and UBB, which are known to be active in cancer cells (Fig. 1c). The top five genes identified to regulate the population of Nestin+ cells included ZNF117, MRLC2, CASS4, DUB3, and ARRB1 (Fig. 1d). Among them, ZNF117 demonstrated the greatest inhibitory effect. Each of the four siRNAs targeting ZNF117 consistently reduced the percentage of Nestin+ cells (Fig. 1d).

### scRNA-seq suggests that ZNF117 regulates GSC differentiation towards oligodendroglial lineage.
To determine which genes regulate GSC differentiation into terminal cell types, we performed scRNA-seq on GS5 cells[15]. To minimize batch-to-batch variability, we prepared four batches of cDNA libraries. Each batch contained between 471 and 1824 cells, 1811–2729 median genes per cell, 5258–9207 median unique molecular identifiers (UMIs), and 20,023–23,000 total genes. In all, we obtained 4592 single-cell transcriptomes at a depth of at least 10,000 reads per cell. After quality-control filtering (see Methods), 3412 cells with expression levels for 21,404 genes were used for downstream analysis (Supplementary Table 2).

To detect specific cell types within the tumor, we visualized GS5 cells with UMAP and grouped them using unbiased, graph-based clustering (Fig. 2a, Supplementary Fig. 2a). Gene expression levels for ZNF117, NES (GSCs), SNAP25 (neurons), GFAP (astrocytes), and PLP1 (oligodendrocytes) were visualized across GS5 cells (Fig. 2b). To identify specific cell populations, we used cell type markers (Supplementary Fig. 2b, c), which had been reported in previous studies to be related to specific cell types[4,14,16–27]. As expected, the same clusters were enriched for the same cell type. In total, we identified 426 GSCs, 802 neuron-like cells, 353 astrocyte-like cells, and 800 oligodendrocyte-like cells from the dataset (Fig. 2c).

Next, we constructed single-cell trajectories from GSCs and differentiated cells to determine which genes mediate GSC differentiation into specific lineages. When aligning the cells in pseudotime, we observed that GSCs settled into the center of the single-cell trajectories, while neuron-like cells, astrocyte-like cells, and oligodendrocyte-like cells were each isolated on their own distinct branches (Fig. 2d). To identify genes that regulate GSC differentiation into each cell type, we determined the differentially expressed genes on each branch, with GSC to neuron-, astrocyte-, and oligodendrocyte-like cells corresponding to states 1, 4, and 3, respectively in Supplementary Fig. 2d.

We compared genes that inhibit GSC differentiation with the 100 genes selected for validation in the RNAi screen and found that 15, 10, and 9 genes overlap as potential candidates which inhibit GSC differentiation towards oligodendroglial (Fig. 2e), neuronal (Supplementary Fig. 2g), and astroglial (Supplementary Fig. 2h) lineages, respectively (Supplementary Table 3). Among the overlapping genes, ZNF117 was significantly differentially expressed in GSC to oligodendroglial differentiation (Fig. 2f), but not other lineages (Supplementary Fig. 2e, f). This finding suggests that ZNF117 may regulate GSC differentiation towards oligodendroglial lineage.

### Experimental validation of ZNF117 as GSC differentiation regulator.
To confirm that ZNF117 regulates GSC differentiation,

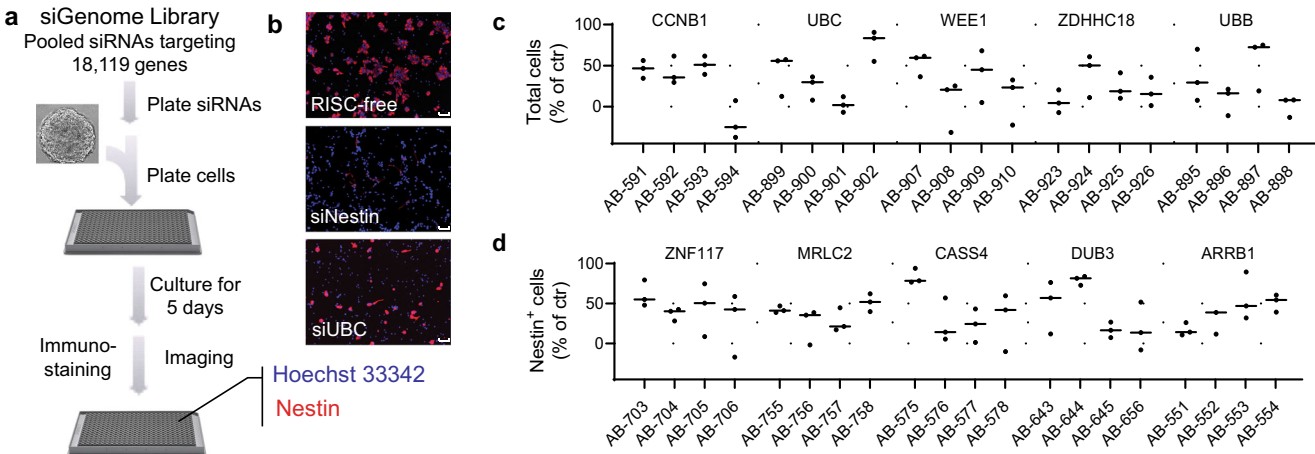

**Fig. 1 Genome-wide RNAi screening identified genes that regulate GSC survival and differentiation. a** Schematic diagram of genome-wide RNAi screening through reverse transfection. Single GS5 cells were plated on polyornithine coating 384-well plates, in which siRNA complexes were pre-assembled. Five days later, cells were stained with Hoechst 33342 (blue, nuclei staining) and anti-Nestin antibody[50], followed by imaging using an Opera high-content screening system. **b** Representative images of cells treated with RISC-free control siRNA, siNestin, or siUBC. Experiments were performed in triplicate. Scale bar, 20 μm. **c** The top 5 genes which demonstrated the greatest inhibitory effects on survival of total cells. **d** The top five genes which demonstrated the greatest effects on reducing Nestin⁺ cell population. For c and d, each gene was targeted by four individual siRNAs, instead of a pool of four siRNAs. AB(AB00477)-xxx represents the ID number of the specific siRNA coded by Dharmacon. Black line represent mean, dots indicate values. Data are presented as mean ± SD ($n = 3$). Statistical differences were determined by two-tailed student's $t$-test.

we differentiated GS5 cells by culturing them in medium containing 10% FBS and examined the changes in expression of *ZNF117*. Western Blot (WB) analysis showed that the expression level of *ZNF117* significantly decreased with culture time; by the end of 3 weeks, when GS5 cells were fully differentiated based on flow cytometry analysis (Supplementary Fig. 3), the expression of *ZNF117* was barely detectable (Fig. 3a).

To further investigate the biological functions of *ZNF117* as a differentiation regulator, we employed CRISPR to downregulate the expression of *ZNF117* in GS5 cells. Unlike siRNA that targets mRNA, Cas9/sgRNAs induces genetic knockout at the genomic DNA level, which allows for the generation of stable cell lines with persistently low expression of the targeted genes. We established GS5 cells with stable overexpression of Cas9 and sgRNAs targeting different regions of *ZNF117* (Supplementary Fig. 4a). Control cells were prepared through the same procedures but with an sgRNA targeting GFP. The resulting cells were designated as GS5sgGFP, GS5sgZNF117-1, and GS5sgZNF117-2. WB analysis showed that the selected sgRNAs efficiently reduced the expression of *ZNF117* in GS5 cells (Fig. 3b). We analyzed the impact of *ZNF117* downregulation on cell proliferation, stemness, and differentiation. Results in Fig. 3c, d show that downregulation of *ZNF117* significantly reduced both cell proliferation rate and stem cell frequency. Further analyses by flow cytometry and immunostaining found that downregulation of *ZNF117* reduced the population of Nestin⁺ cells, while notably increasing the GalC⁺, but not GFAP⁺, or Tuj-1⁺, population within GS5 cultures (Fig. 3e–g, Supplementary Fig. 4b). The increase in oligodendroglial population was further confirmed by the expression of OLIG1 (Supplementary Fig. 4c). Next, we overexpressed *ZNF117* gene in GS5sgZNF117-1 cells and found the phenotypic changes associated with downregulation of *ZNF117* were reversed (Supplementary Fig. 5). These findings, which are consistent with the observations from both the RNAi screen (Fig. 1d) and scRNA-seq analysis (Fig. 2c–e), suggest that *ZNF117* regulates GSC differentiation towards the oligodendroglial lineage.

We determined the effect of *ZNF117*-induced differentiation on tumor development in vivo through inoculation of engineered cells into the brains of nude mice. Control mice received injection of the same amount of GS5sgGFP cells. Afterward, the mice were

monitored for survival. We found that downregulation of *ZNF117* significantly prolonged the survival of tumor-bearing mice. Mice inoculated with GS5sgZNF117-1 cells, which have the lowest expression level of *ZNF117* among all the three cells, survived for >100 days post-inoculation. In contrast, the median survival for mice bearing tumors derived from GS5sgZNF117-2 cells was 49 days, compared to 37 days for mice bearing GS5sgGFP tumors (Fig. 3h). The observed difference in inhibition of tumor formation between the two sgRNAs could be attributed to the fact that, compared to sgZNF117-2, sgZNF117-1 is more efficient in down-regulating *ZNF117* expression (Fig. 3b) and in inducing differentiation (Fig. 3e–g). To exclude the possibility that the marked tumor-inhibitory ability of sgZNF117-1 is caused by off-target effects, we identified genes that have mismatches with sgZNF117-1 and determined whether these genes have detectable disruption after sgZNF117-1-was delivered. No genes had zero or one mismatch with sgZNF117-1, and six genes had two to four mismatches in exons. We sequenced the six genes and did not detect significant cutting activity (Supplementary Fig. 6). H&E staining showed that loss of sgZNF117 correlated with dramatically decreased tumor size (Fig. 3i). Ki-67 staining revealed that residual tumors derived from GS5sgZNF117-1 cells had reduced proliferation rates (Supplementary Fig. 4d).

Collectively, these results suggest that *ZNF117* regulates GS5 cell differentiation towards oligodendroglial lineage, and down-regulation of *ZNF117* inhibits the proliferation, stemness, and tumor development of GSCs. Compared to sgZNF117-2, sgZNF117-1 reduced the expression of *ZNF117* at higher efficiency and led to greater differentiation effects. Therefore, sgZNF117-1 was selected for further studies and is designated sgZNF117 in the remainder of the manuscript.

**Validation in additional GSC cell cultures and xenografts.** To exclude the possibility that the observed differentiation effects of *ZNF117* is limited to GS5 cells, we applied the same approach to characterize the role *ZNF117* in PS30 and PS24 cells, two additional well-characterized GSC cultures[6,15]. We found that the expression of *ZNF117* in PS30 cells decreased with continuous culture in FBS-containing medium (Fig. 4a). Through transduction with lentiviruses containing sgZNF117, we generated PS30

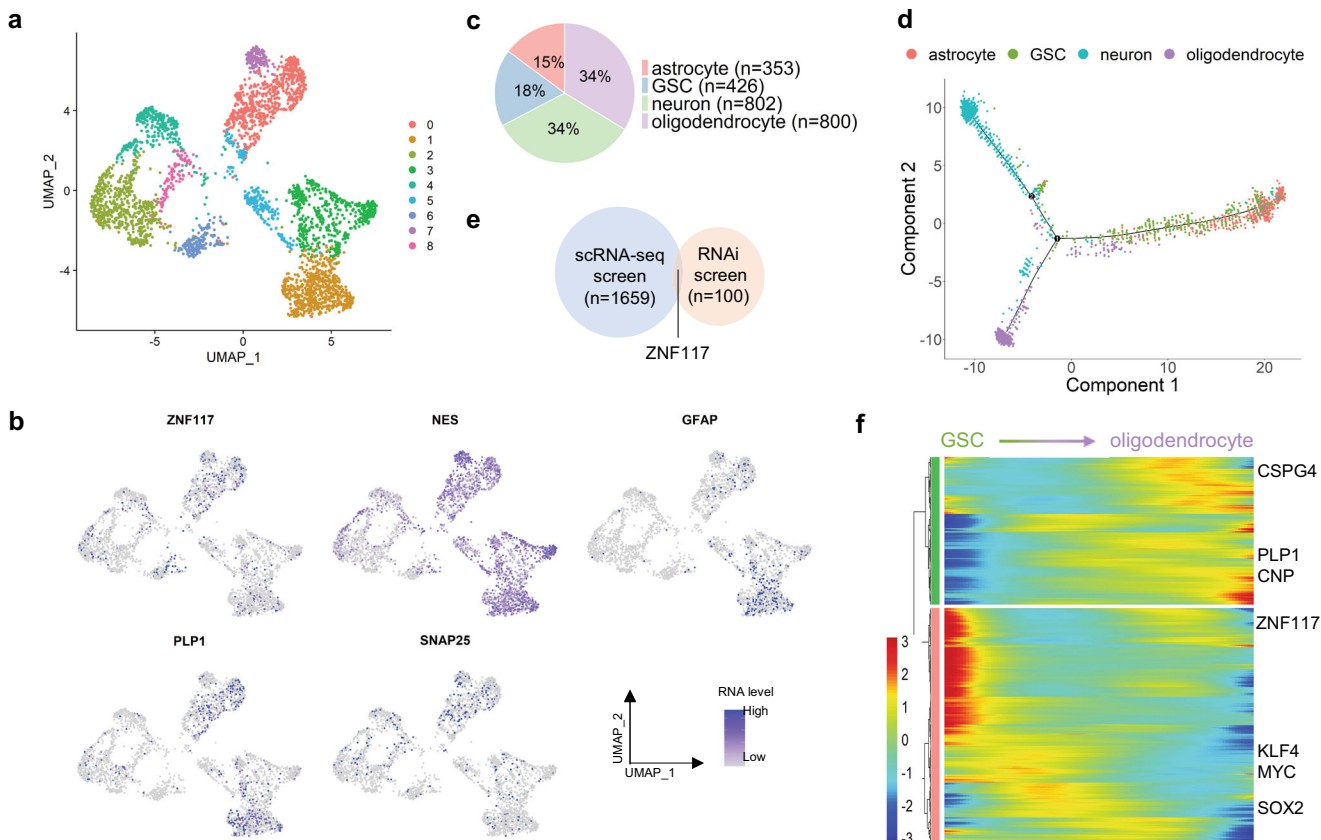

**Fig. 2 Single-cell trajectory analysis suggests that *ZNF117* regulates GSC differentiation towards an oligodendroglial lineage. a** UMAP of scRNA-seq data from GS5 cells. Unsupervised clustering based on top 1000 variable genes reveals cell–cell heterogeneity within the GS5 tumor population. **b** Expression of cell type-specific markers and *ZNF117* in UMAP. Expression of *NES* (GSC), *GFAP* (astrocytes), *PLP1* (oligodendrocytes), and *SNAP25* (neurons) is shown. **c** Distribution of cell types identified from GS5 scRNA-seq data. Neurons and oligodendrocyte each compose of around 33% of the tumor, while GSCs and astrocytes each compose of 18% and 15% of the tumor, respectively. **d** Single-cell trajectories of cell types identified in **c**. GSCs are in the center of the graph, while each of the differentiated cell types form their own distinct branch. **e** Venn diagram of significant genes that regulate oligodendrocyte differentiation from single-cell trajectory analysis and Nestin expression from the RNAi screen. *ZNF117* appears in both screens. **f** Heatmap of significant genes that regulate GSC to oligodendrocyte differentiation. *ZNF117* decreases in expression as GSCs differentiate into oligodendrocytes. Color bar represents normalized expression. Genes are hierarchically clustered according to their expression pattern. *q*-value < 5e-02.

cells with downregulation of *ZNF117* (Fig. 4a). The resulting PS30^sgZNF117 cells were characterized for cell proliferation, stemness, differentiation, and tumorigenicity. Cells treated with sgGFP (PS30^sgGFP cells) were included as a control. We found that downregulation of *ZNF117* effectively reduced the rate of cell proliferation (Fig. 4b) and frequency of stem cells (Fig. 4c), induced differentiation preferentially towards oligodendroglial lineage (Fig. 4d–f), and inhibited tumor development in mice (Fig. 4g, *P* = 0.0007). The same trends were also observed in PS24 cells in vitro and in vivo (Supplementary Fig. 7). Therefore, our findings in both GSC cultures and mouse xenografts are consistent with those identified in GS5 cells, suggesting that the biological effects of *ZNF117* are not unique to GSCs of specific origin.

***ZNF117* regulates Notch signaling through interaction with *JAG2*.** The *ZNF117* gene encodes a protein containing multiple C2H2-type zinc finger motifs and is predicted to have DNA-binding transcription factor activity based on gene ontology annotations. To determine the transcriptional activity of *ZNF117*, we performed a whole-transcript expression analysis using Affymetrix HuGene-2.0 arrays and found GS5^sgZNF117 and control GS5^sgGFP cells to be markedly different at the transcriptional level (Fig. 5a). In total, 3,642 transcripts were differentially expressed by over 2-fold (Supplementary Data 2), including 1781

that were up-regulated and 1861 that were downregulated in GS5^sgZNF117-1 cells. Pathway analysis of both the up-regulated and downregulated genes reveals that the Notch pathway is among the major pathways which are potentially regulated by *ZNF117* (Supplementary Fig. 8a). Further analysis revealed that major Notch-related genes, including *NOTCH1*, *NOTCH2*, *NOTCH3*, *DLL3*, *DTX1*, *DTX3*, *LFNG*, *MAML2*, and *NOTCH2N*, are in the downregulated cohort and, *NOTCH1*, *MET*, and *PRDX1*, are in the up-regulated cohort (Fig. 5b). Down-regulation of *NOTCH1*, *NOTCH2*, and *NOTCH3* was confirmed by qRT-PCR analysis (Supplementary Fig. 8b). We further quantified the activity of the Notch signaling using Cignal™ pathway reporters and found that downregulation of *ZNF117* significantly inhibited Notch activity (Fig. 5c). No significant differences in activity of Wnt signaling were detected between GS5^sgZNF117 and control GS5^sgGFP cells (Supplementary Fig. 8c). These results suggest that *ZNF117* exerts its biological functions through regulation of the Notch pathway.

To identify specific genes which are regulated by *ZNF117*, we performed ChIP-seq. In total, DNA fragments of 4,616 genes were enriched. Among them, 8%, 11%, and 81% were located in introgenic, proximal, and intergenic regions, respectively (Supplementary Fig. 8d). For 327 genes, including *JAG2*, the DNA fragments were located in the proximal promoter region (Fig. 5d, e, Supplementary Data 3). We examined three genes, including

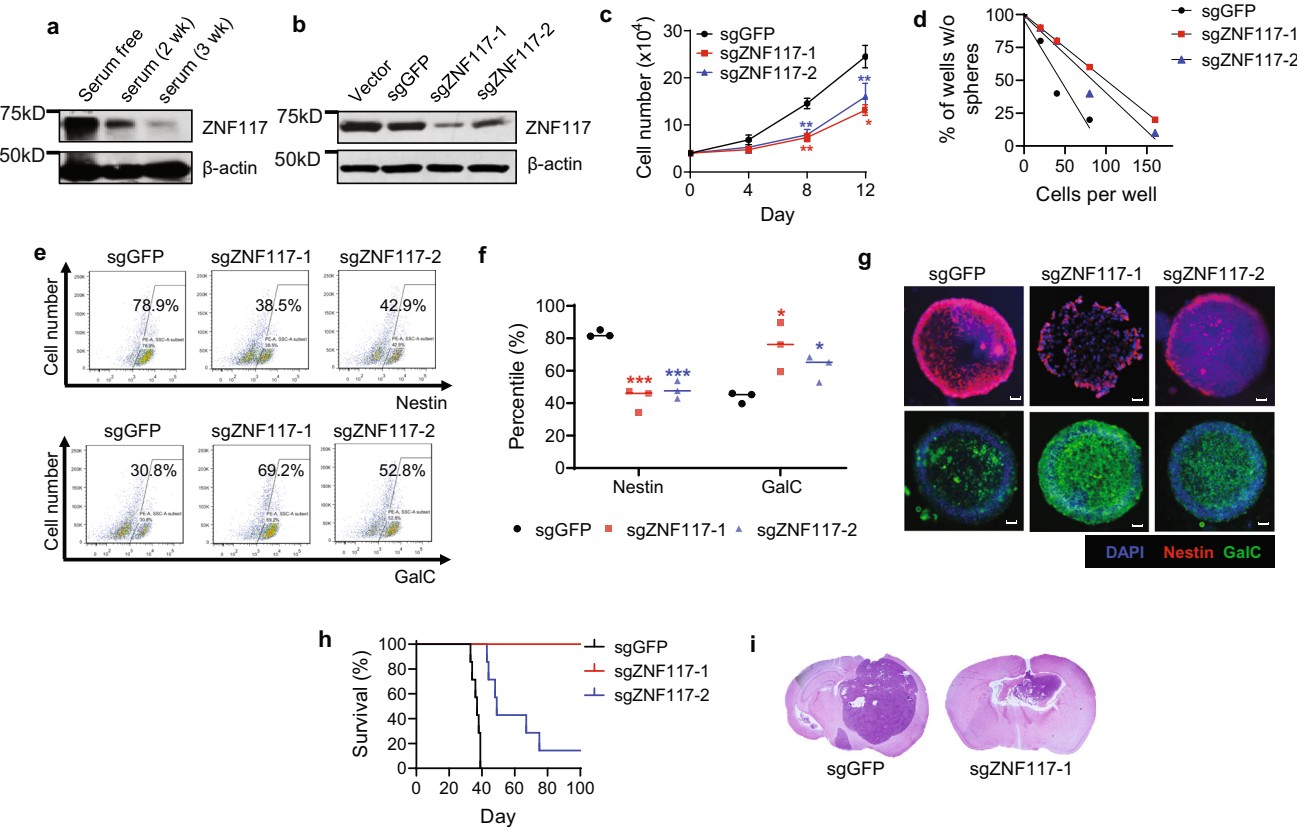

**Fig. 3 *ZNF117* regulates GSC differentiation towards oligodendroglial lineage. a** WB analysis of the expression of *ZNF117* in GS5 cells cultured in serum-free or cultured in serum-containing medium for 2 or 3 weeks (wk). Experiments were performed in triplicate. **b** WB analysis of the expression of *ZNF117* in GS5 parental cells and cells treated with Cas9 and the indicated sgRNAs. **c** Proliferation of GS5 cells treated with Cas9 and the indicated sgRNAs ($n = 3$ biologically independent samples). **d** Stem cell frequency of GS5 cells treated with Cas9 and the indicated sgRNAs as determined by limiting dilution assay. **e–g** Representative flow cytometry images (**e**) and corresponding quantification (**f**) ($n = 3$ biologically independent samples, black line represents mean, dots indicate values), and immunostaining (**g**) of Nestin+ and GalC+ cell populations in GS5 cells after treatment with Cas9 and the indicated sgRNAs. **h** Kaplan–Meier survival analysis of mice received inoculation of the indicated cells. **i** H&E staining of residual tumors isolated from mice receiving inoculation of the indicated cells. Data are presented as mean ± SD ($n = 3$). Statistical differences were determined by two-tailed student's *t*-test. *$P$-value < 0.05; **$P$-value < 0.01; ***$P$-value < 0.001. Scale bar, 50 μm.

*IL6ST*, *CDH4*, and *JAG2*, which are known to be relevant to GBM, by ChIP-PCR, and confirmed that all of them were enriched (Fig. 5f, Supplementary Fig. 8e). Because the cDNA array analysis found that Notch signaling is regulated by *ZNF117*, we focused on *JAG2*, a Notch ligand known to regulate the Notch pathway[28,29]. We found that downregulation of *ZNF117* reduces the expression of *JAG2* (Fig. 5g). To determine how *ZNF117* directly regulates *JAG2*, we cloned sequences of 500, 1500, and 7150 flanking the predicted *ZNF117*-binding site within *JAG2* into PGL4.2-TATA luciferase construct (Fig. 5h) and determined the expression of luciferase in cells that were engineered to express either Cas9/sgGFP or Cas9/sgZNF117 after transfection with the reporter constructs. We found that, compared to those in control cells, the luciferase signal in Cas9/sgZNF117 expressing cells was significantly lower (Fig. 5i). To further provide evidence that *ZNF117* regulates GSC differentiation through *JAG2*, we overexpressed *JAG2* under a CMV promoter in GS5^sgZNF117-1 cells and found that overexpression reversed *ZNF117* knockout-mediated oligodendroglial lineage differentiation (Supplementary Fig. 5). Taken together, these results suggest that *ZNF117* regulates Notch signaling through interaction with *JAG2*.

**ZNF117 as a therapeutic target for GBM differentiation therapy.** Analysis of RNA-seq database by The Cancer Genome Atlas

(TCGA) shows that *ZNF117* is expressed at a higher level in GBM than normal brain tissues (Supplementary Fig. 9a). Further analysis of several GBM public databases, including TCGA RNA-seq database, The Repository of Molecular Brain Neoplasia Data (REMBRANDT) database, and Chinese Glioma Genome Atlas (CGGA), found that the expression of *ZNF117* is negatively correlated with patient survival (Supplementary Fig. 9b–d). Collectively, these analyses suggest *ZNF117* as a potential therapeutic target for GBM differentiation therapy.

We assessed if *ZNF117* can be targeted for GBM treatment experimentally in tumor-bearing mice. To enable downregulation of *ZNF117* in vivo, we employed liposome-templated hydrogel nanoparticles (LHNPs), which we recently developed for targeted delivery of CRISPR/Cas9 to brain tumors[30]. LHNPs were synthesized with encapsulation of ribonucleoprotein (RNP) complexes of Cas9 protein and sgRNA targeting either *ZNF117* or GFP. The NPs were further engineered through surface conjugation of iRGD, a peptide with high affinity for $\alpha_v\beta_3/\alpha_v\beta_5$ integrins[31,32], and internal encapsulation of Lexiscan, a small molecule that can transiently open the blood-brain barrier (BBB)[33,34] (Fig. 6a). As a result, LHNPs can penetrate the BBB and accumulate preferentially in tumors through an autocatalytic, brain tumor-targeting mechanism[35–37]. The resulting LHNPs are spherical in shape, have a diameter of ~90 nm (Fig. 6b), and were confirmed to selectively transfect

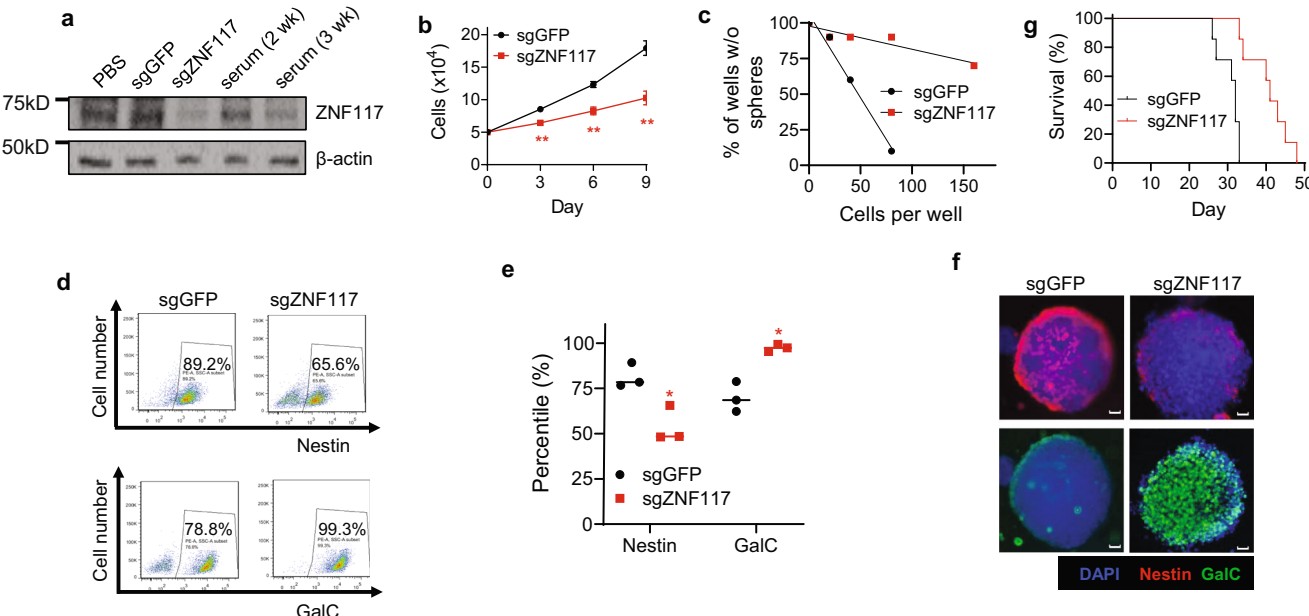

**Fig. 4 Validation of the differentiation effect of *ZNF117* in PS30 cells. a** WB analysis of the expression of *ZNF117* in PS30 cells cultured in serum-containing medium or with constitutive expression of Cas9 and the indicated sgRNAs. Experiments were performed in duplicate. **b**, **c** Proliferation (**b**) and stem cell frequency (**c**) of PS30 cells treated with Cas9 and the indicated sgRNAs ($n = 3$ biologically independent samples). **d–f** Representative flow cytometry images (**d**) and corresponding quantification ($n = 3$ biologically independent samples, black line represents mean, dots indicate values). **e** and immunostaining (**f**) of Nestin$^+$ and GalC$^+$ cell populations in PS30 cells after treatment with Cas9 and the indicated sgRNAs. Scale bar, 50 μm. **g** Kaplan–Meier survival analysis of the mice received inoculation of the indicated cells ($n = 7$). Data are presented as mean ± SD. Statistical differences were determined by two-tailed student's $t$-test. \*$P$-value < 0.05; \*\*$P$-value < 0.01.

brain tumor cells after intravenous administration (Supplementary Fig. 10). We characterized LHNPs in GS5 and PS30 cells. We found that the expression of *ZNF117* in both PS30 and GS5 cells treated with LHNPs loaded with *ZNF117*-targeted RNP was significantly decreased, compared to control cells treated with LHNPs loaded with GFP-targeted RNP (Fig. 6c, Supplementary Fig. 11a). Because *ZNF117* expression decreased more significantly in PS30 cells compared to GS5 cells after treatment, we chose to determine the therapeutic benefit of *ZNF117*-targeted therapy in PS30-derived mouse xenografts. Mouse xenografts were established through intracranial inoculation of PS30 cells that were engineered to express luciferase. Seven days later, the mice were randomly grouped and treated with LHNPs loaded with *ZNF117*-targeted RNP, PBS, or control LHNPs loaded with GFP-targeted RNP. Treatments were given at a dose of 1 mg NPs (sgRNA equivalent dose of 0.2 ug) per injection intravenously through tail veins three times a week for three consecutive weeks. The mice were monitored for survival and tumor development, which was determined based on luciferase imaging by an In Vivo Imaging System (IVIS). Results in Fig. 6d, e show that treatment with LHNPs loaded with *ZNF117*-targeted RNP significantly inhibited tumor growth and improved the survival of tumor-bearing mice. The median survival of the treatment group was 48 days, compared to 33 days for mice treated with control LHNPs ($P = 0.007$) and 32 days for mice treated with PBS. At the end of the study, the mice were euthanized, and their brains were isolated and examined. H&E staining confirmed that residual tumors in control groups were significantly larger than those in the treatment group; in contrast to tumors in the control groups, the residual tumors in the treatment group showed a benign phenotype with reduced nuclear-to-cytoplasm ratios (Fig. 6f). Further WB analysis showed that compared to that in the

control group, the expression of *ZNF117* in residual tumors isolated from the treatment group was significantly lower (Fig. 6g). Consistent with the findings in cell culture (Fig. 4d, e), immunostaining revealed that treatment with LHNPs loaded with *ZNF117*-targeted RNP significantly reduced Nestin$^+$ while enriching GalC$^+$ cells (Fig. 6h). These findings suggest that the observed inhibition of tumor development in the treatment group resulted from *ZNF117*-mediated differentiation.

Previous studies suggest that GSCs, but not differentiated cells, are resistant to TMZ, and improved treatment of GBM can be achieved through combination of TMZ chemotherapy and differentiation therapy[3,38]. Consistently, we found that down-regulation of *ZNF117* sensitized both PS30 and GS5 cells to TMZ treatment in vitro (Fig. 6i, Supplementary Fig. 11b). We determined the efficacy of TMZ chemotherapy in combination with *ZNF117*-mediated differentiation therapy in tumor-bearing mice. Seven days after inoculation of PS30 cells, mice were treated with TMZ alone or TMZ in combination with LHNPs loaded with *ZNF117*-targeted RNP. The same treatment regimen as described above was used. TMZ was given at 0.1 mg/kg intraperitoneally after intravenous administration of LHNPs. We found that treatment with LHNPs loaded with *ZNF117*-targeted RNP significantly prolonged the survival of mice receiving treatment with LHNPs loaded with GFP-targeted RNP (41 days vs. 33 days, $P = 0.007$). We found that the combination therapy resulted in greater therapeutic benefits than either TMZ or LHNPs alone (Fig. 6d–f). The median survival of mice in the combination group was 48 days; in comparison, the median survival for the TMZ treatment group and the group treated with LHNPs loaded with *ZNF117*-targeted RNP were 37 days ($P = 0.004$) and 42 days ($P = 0.02$), respectively. The results suggest that *ZNF117*-mediated GSC differentiation therapy sensitizes GBM to TMZ chemotherapy.

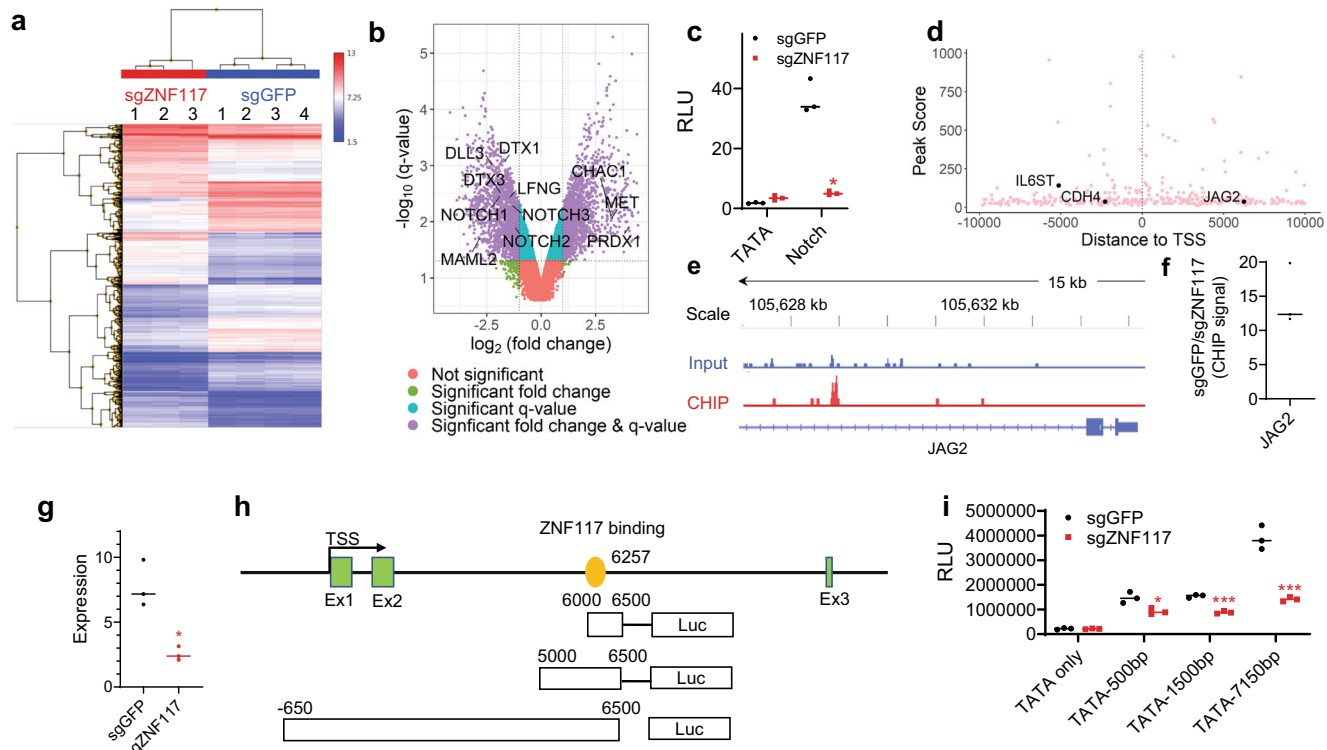

**Fig. 5 ZNF117 regulates Notch signaling through interaction with JAG2. a** Heatmap of genome-wide transcription profiles of GS5^sgGFP and GS5^sgZNF117-1. Colors represent normalized expression. **b** Volcano plot shows that *ZNF117* regulates Notch signaling. **c** Pathway reporter assay suggests that Notch activity is downregulated in *ZNF117* knockout cells. **d** Genes identified by CHIP-seq and their distance to transcription start site (TSS). **e** Genome browser view of the distribution of *ZNF117*-binding regions identified by ChIP-seq. **f** ChIP-PCR validation of *JAG2* as a *ZNF117*-binding candidate. **g** Expression of *JAG2* in GS5^sgGFP and GS5^sgZNF117 as determined by qRT-PCR. **h** Schematic diagram of *JAG2* reporter construction. **i** Characterization of the effect of *ZNF117* downregulation on transcription activity of the indicated sequences within *JAG2* based on luciferase reporter assay. Data are presented as mean ± SD. Black line represents mean, dots indicate values (**c**, **f**, **g**, **i**). Statistical differences were determined by two-tailed student's *t*-test. *P-value < 0.05; **P-value < 0.01; ***P-value < 0.001.

## Discussion

Differentiation therapy has great potential to improve the treatment of GBM[2–5,7]. In this study, we identified *ZNF117* as a genetic regulator which controls differentiation of GSCs towards oligodendroglial lineage. We demonstrate that *ZNF117* can be targeted for GBM treatment alone or in combination with TMZ. Mechanistically, we show that *ZNF117* interacts with *JAG2* and regulates Notch signaling, which is known to be critical for GSC self-renewal and differentiation[39,40] (Supplementary Fig. 12). The exact mechanism for how *ZNF117* regulates GSC differentiation towards oligodendroglial lineage has yet to be investigated.

Differentiation of GSCs has been previously explored. Previous studies showed that GSCs can be induced differentiation through treatment with bone morphogenetic protein 4 (BMP-4)[38], or through downregulating expression of *TRRAP* using siRNA[8]. Both *BMP4* and *TRRAP* regulate GSC differentiation towards an astroglial lineage. As GSCs possess stem cell properties and are able to differentiate towards neuronal and oligodendroglial lineages in addition to the astroglial lineage[14], identification of differentiation genes solely based on GFAP expression excludes those regulating differentiation towards neuronal or oligodendroglial lineage. We show that a combination of genome-wide RNAi screening with scRNA-seq allows for the identification of genes which regulate GSC differentiation to all three lineages.

Despite its significance, there are limitations associated with this study. First, GBM is known to be highly heterogeneous. Although the function of *ZNF117* was experimentally characterized in three GSC cultures (Figs. 3, 4, Supplementary Fig. 7), it is unknown whether *ZNF117* regulates GSC differentiation across other GBM

samples in the same manner. Second, we found that, similar to astrocytic differentiation[8,38]; oligodendroglial differentiation sensitizes GSC to TMZ treatment (Fig. 6). However, the relevance of the two different differentiation lineages in the context of therapeutic development has yet to be determined. This finding also implicates that improved treatment of GBM can be achieved through targeting differentiation pathways other than astrocytic lineage. Lastly, *ZNF117* functions as a transcription factor, which is often considered an "undruggable" target. Translation of *ZNF117*-targeted therapy through the use of traditional therapeutics is a major challenge, and further studies are required.

In summary, our study reveals *ZNF117* as a major GSC differentiation regulator which can be potentially targeted for GBM treatment. Our study is also significant in that it suggests a direction to characterize GSC differentiation through genome-wide functional screening in combination with scRNA-seq.

## Methods

**Cell culture and materials**. HEK293T cells were purchased from the ATCC. GS5 cells were kindly provided by the Lamszus lab[14]. PS30 and PS24 cell cultures were established at Yale with the approval by the appropriate Institutional Review Boards as we previously reported[6,15]. All cells were cultured in Neurobasal-A media (Invitrogen) supplemented with B27 nutritional supplement (Gibco), basic fibroblast growth factor (bFGF), and epidermal growth factor (EGF) unless otherwise noted. Details about the antibodies used in this study are described in Supplementary Table 4.

**Genome-wide RNAi screen**. The screen was carried out using the siGenome human genome library from Dharmacon (GE Healthcare) through reverse transfection, and each gene was tested separately. Briefly, siRNA in the black 384-well PerkinElmer Cell Carrier plates was dissolved in OptiMEM medium (Thermo Fisher Scientific). After brief centrifugation, the plates were incubated at room

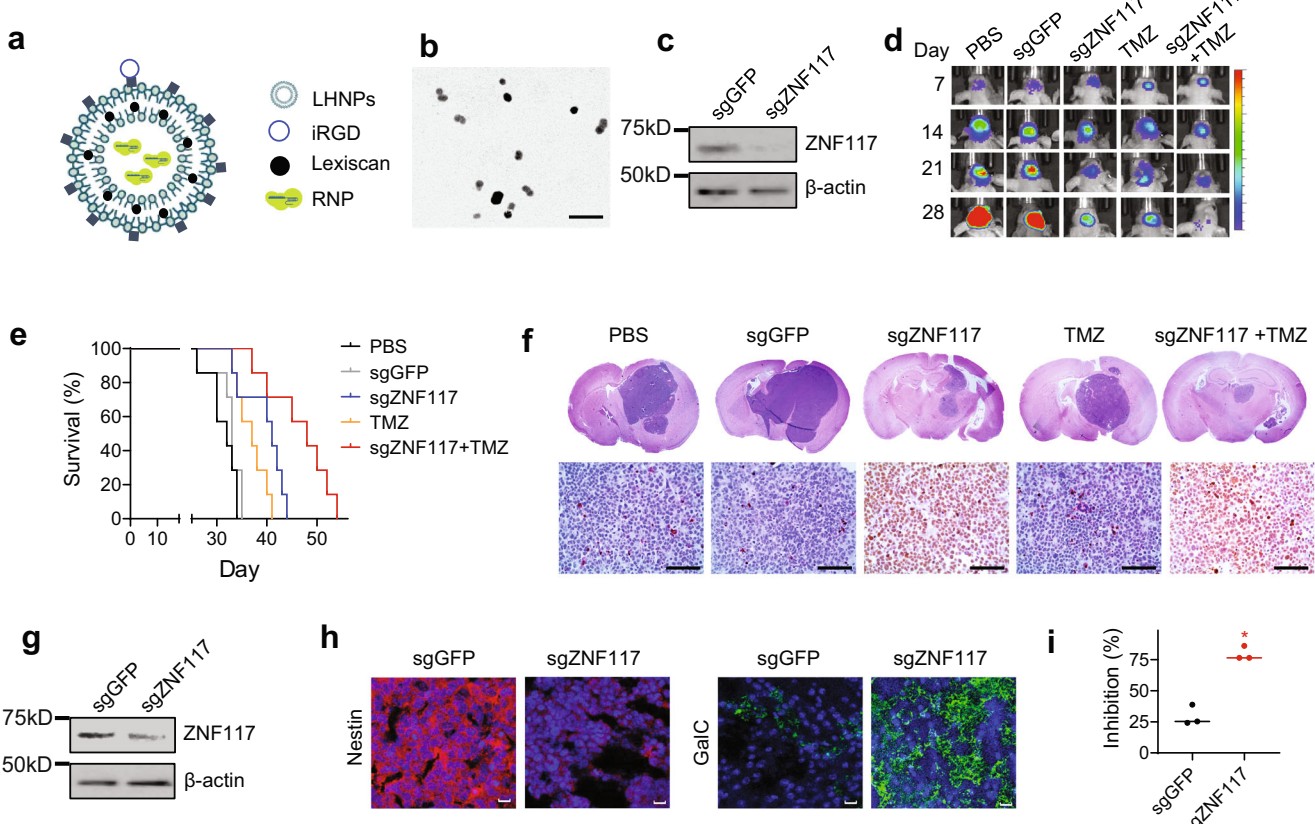

**Fig. 6 Characterization of *ZNF117* as a therapeutic target for GBM differentiation therapy. a** Schematics of LHNPs. **b** A representative image of LHNPs as captured by transmission electron microscopy (TEM). Experiments were performed in triplicate. Scale bar: 500 nm. **c** WB analysis of *ZNF117* expression in PS30 cells treated with LHNPs loaded with Cas9 and sgGFP or sgZNF117. Experiments were performed in triplicate. **d** Representative images of tumors in the brain based on IVIS imaging. **e** Kaplan–Meier survival analysis of mice receiving the indicated treatments. **f** Representative H&E images of the brain isolated from mice receiving the indicated treatments. Experiments were performed in triplicate. Scale bar, 50 μm. **g** WB analysis of *ZNF117* expression in residual tumors isolated from mice received the indicated treatments. Experiments were performed in triplicate. **h** Representative immunostaining images of residual tumors isolated from mice received the indicated treatments for expression of Nestin and GalC. Scale bar, 20 μm. **i** Inhibition of the proliferation of PS30 cells treated with Cas9 with the indicated sgRNA by TMZ ($n = 3$ biologically independent samples). Data are presented as mean ± SD. Black line represents mean, dots indicate values. Statistical differences were determined by two-tailed student's *t*-test. *P-value < 0.05.

temperature (RT) for 5 min, following by adding Lipofectamine RNAimax (Thermo Fisher Scientific). Five minutes later, 5 μl of OptiMEM/Lipofectamine mix was added to the plates, followed by 20 min incubation at RT. Single GS5 cells were prepared using acutase and resuspended in Neurobasal-A media with growth factors to a final concentration of 2000 cells/well. Ten microliters of cell suspension were then added to the plates. Five days later, the plates were fixed by formaldehyde. The cells were then permeabilized and stained with primary antibodies against Nestin (Sigma) and secondary antibodies Alexa Fluor® 546 (Thermo Fisher Scientific). After washing with PBS, the cells were incubated with 1 μg/ml Hoechst (Sigma) for 15 min at RT, followed by a final wash in PBS. Throughout the procedures, a Multidrop Combi Reagent dispenser (Thermo Fisher Scientific) was used to ensure even liquid addition. Plates were then imaged using the PerkinElmer Opera high-content confocal screening platform. Thirty views were captured for each well using a 20x air objective, numerical aperture. Images were analyzed using custom workflows created with the Columbus software (PerkinElmer). As a primary readout, the average center intensity was determined. The primary screen was analyzed based on the robust z (RZ)-score. For the secondary confirmation screen, 100 genes were selected based on their ability to substantially decrease in cell number (>50% reduction in cell number in at least 2/3 replicates) or significantly reduce Nestin+ population (RZ < −2) after downregulation. The secondary library siRNA plates with individual siRNAs were used to confirm the selected candidates. The confirmation experiments were carried out using the same procedure.

**scRNA-seq analysis.** scRNA-seq was performed using a microchip-based platform, scFTD-seq, which we recently developed[41]. GS5 cells were sequenced using 75 bp pair-end reads on a HiSeq2500 instrument (Illumina) in HighOutput Mode V4. Raw reads were preprocessed for cell barcodes and UMIs, and then aligned to the human genome(hg19) using STAR v2.5.2b[42]. A digital gene expression matrix was generated for each cell with over 10,000 reads.

Seurat[43] (V2.3.0) in R (V3.6.2) was used to analyze the digital expression matrix generated from the GS5 cells. Cells that met quality control conditions (unique number of genes between 1000 and 5000 and <10% of genes are mitochondrial genes per cell) were included for downstream analysis. Three thousand four hundred twelve cells with expression levels for 21,404 genes passed quality control. Genes that were differentially expressed in each cluster were identified using the Seurat function Find Markers, which returns the gene names, average log fold-change, and adjusted p-value for genes enriched in each cluster. Unsupervised clustering was performed to identify clusters of cells based on cell type and then visualized using *t*-SNE.

Monocle[44–46] (V2.14.0) was used to plot single-cell trajectories of GS5 cells over pseudotime to visualize GSC differentiation. A Seurat object containing GSCs, neurons, astrocytes, and oligodendrocyte identified from the GS5 dataset was imported into Monocle using the importCDS() function. Reversed graph embedding was performed in an unsupervised manner to identify differentially expressed genes. Cells were then ordered in pseudotime with GSCs defined as the root state. Because each terminal cell type was isolated on its own branch, genes that regulate GSC differentiation for a specific cell type were visualized by following a specific state of the trajectory. Differential genes for each state were visualized and identified by the plot_multiple_branches_pseudotime() function. Significant genes for the scRNA-seq single-cell trajectory screen were defined as genes that were in the cluster with high expression in the GSC state and low expression in the differentiated state, mean gene expression >0.14 mRNA transcripts per cell, maximum expression of at least three mRNA transcripts per cell, and non-mitochondrial or ribosomal genes.

**Constructs, transfection, and retroviral transduction.** lentiCas9-Blast (Addgene #52962) and lentiGuide-Puro (Addgene #52963) plasmids are gifts from Feng Zhang[47]. Sequences for the sgRNAs using in this study include: sgZNF117-1:

TGTAGAAATTCACTCTAGTT; sgZNF117-2: GTTGAGTGTAAACAGCACAA; and sgGFP: GAGCTGGACGGCGACGTAAA. These oligos were synthesized by the Keck Biotechnology Resource Laboratory at Yale and cloned into lentiGuide-Puro. Production and transduction of artificial lentivirus were carried out according to our previously published procedures[37].

**Synthesis and characterization of LHNPs**. LHNPs were synthesized according to our recently published procedures[36]. Briefly, RNP was formed by complexing Cas9 protein with candidate sgRNA that was synthesized at Synthego with chemical modifications, mixed with PEI-adamantine (AD) and PEI-CD, and encapsulated into liposomes consisting of 1,2-dioleoyl-3-trimethyl-lammonium-propane chloride salt (DOTAP, Avanti Polar Lipids), cholesterol (Avanti Polar Lipids), and DSPE-PEG$_{2000}$-MAL (Avanti Polar Lipids) through extrusion. Morphology of LHNPs was determined using a TEM microscope (FEI Tecnai TF20 TEM).

**Characterization of cytotoxicity**. Cells were plated in 96-well plates at a density of 4000 cells in 200 μL per well and treated with TMZ s at 25uM after overnight incubation. After 72 h, cell proliferation was quantified using the standard dimethyl thiazolyl diphenyl tetrazolium salt (MTT) assay. The absorption, which is proportional to the number of live cells, was measured at 570 nm using a BioTek Instrument ELx800 microplate reader.

**Sphere formation assay and limiting dilution assay**. Sphere formation assay was carried out according to our previously published procedures[48]. Briefly, cells were seeded in 96-well plates at a density of 1000 cells/well and maintained in the Neurobasal medium supplemented with growth factors at 37 °C. Tumor sphere sizes and sphere numbers were assessed using Nikon microscope 7 days after implantation. For limiting dilution assays, cells were plated in 96-well plates at 1, 5, 10, 20, 40, or 80 cells per well, with ten replicates for each cell number. The presence of spheres in each well was determined after 7 days of culture. Limiting dilution analysis was performed using online software (http://bioinf.wehi.edu.au/software/elda/).

**Gene expression profile assay**. Whole-transcript expression analysis was performed using Affymetrix HuGene-2.0 arrays. Analyses were performed according to our previous report[37,48]. Briefly, 2 weeks after antibiotic selection, control sgGFP-treated GS5 cells (four biological replicates) and sgZNF117-treated GS5 cells were collected. Total RNA was isolated using the RNeasy Mini Kit (QIAGEN). Five micrograms of total RNA from each sample was sent to the Yale Center for Genome Analysis for expression microarray profiling. Batch variations were adjusted by Affymetrix Transcriptome Analysis Console (TAC) Software. Genes with >2-fold expression changes and a significance threshold of FDR-adjusted ($P < 0.05$) were subjected to pathway analysis using the IPA (Ingenuity Systems; https://digitalinsights.qiagen.com/). Gene Ontology analysis was performed using PANTHER[49].

**ChIP-sequencing assay and ChIP-qPCR**. Native chromatin immunoprecipitation (N-ChIP) assay was performed using an EZ-Magna ChIP™ A/G Chromatin Immunoprecipitation Kit (EMD Millipore) according to the manufacture's instructions. Ten million GS5 cells were used for each ChIP and massive parallel sequencing (ChIP-seq) experiment. Cell fractionation and chromatin pellet isolation were performed. Chromatin pellets were briefly digested with micrococcal nuclease, and the mononucleosomes were monitored by electrophoresis. Co-purified DNA molecules were isolated and quantified (100–200 ng for sequencing). Co-purified DNA and whole-cell extraction (WCE) input genomic DNA were subject to library construction, cluster generation, and next-generation sequencing (Illumina HiSeq 2000). Peaks were called by Model-based Analysis of ChIP-Seq (MACS) V1.3 and were visualized by Integrative Genomics Viewer (IGV) v1.5. ChIP-qPCR primers are listed in Supplementary Table 5.

**qRT-PCR**. Total RNA was extracted from cells using TRIzol (Invitrogen) according to the manufacturer's instructions. cDNA synthesis was performed with 500 ng of total RNA with miRScript II RT KIT (QIAGEN) and subjected to SYBR green (Applied Biosystems). Real-time qPCR was performed according to the manufacturer's protocol using CFX96 Real-Time System (Bio-Rad). Relative expressions were calculated by $2^{-\Delta CT}$. qRT-PCR primers are listed in Supplementary Table 5.

**Western blot and immunohistochemistry**. For WB analysis, total protein from cells or tissue was extracted using RIPA buffer (Cell Signaling, USA) supplemented with proteinase inhibitors cocktail tablets (Roche, Catalog # 04693116001). WB analysis was carried out according to the standard protocol. Briefly, 10–20 μg of protein lysates were first separated using 6–12% SDS-PAGE. Afterward, the protein was transferred to PVDF membranes (EMD Millipore, USA), which were then incubated with the selected primary antibodies and then secondary antibodies. Expression of candidate proteins were detected by an ECL kit (Pierce, USA). For immunohistochemistry analysis, sections of tissue or cells were prepared, fixed, and stained using primary antibodies. After extensive wash, the tissue or cells were incubated with the Fluor or HRP labeled secondary antibodies, and then colorized with the avidin-biotin-peroxidase complex (ABC) method following the manufacturer's protocol. Images were acquired using a laser scanning confocal microscope FV1000 (Olympus, Japan).

**Flow cytometry**. GBM cells were dissociated into single cells using Accutase (Sigma). For each staining, 1 million cells were fixed by 4% PFA for 15 min at room temperature and then permeabilized by 1x Perm/Wash (BD) buffer for 30 min. Cells were incubated with a primary antibody for 1 h in room temperature, followed by incubation with a secondary antibody for 15 min. All cells were analyzed on a Becton Dickinson LSRII flow cytometer using Flowjo software. An example gating strategy is provided in Supplementary Fig. 13.

**Animal studies**. This study was compliant with all relevant ethical regulations regarding animal research and was approved by the Institutional Animal Care and Use Committee of Yale University. Female athymic nude mice (BALB/c nu/nu, ~6 weeks old) were purchased from Charles River Laboratories and maintained at Yale Animal Resource Center. Tumor inoculation was performed according to our published procedures[35]. Briefly, mice were anesthetized via intraperitoneal injection of ketamine and xylazine. About $5 \times 10^4$ cells in 5 μL of PBS were injected into the right striatum 2 mm lateral and 0.5 mm posterior to the bregma and 3 mm below the dura using a stereotactic apparatus with an UltraMicroPump (UMP3) (World Precision Instruments, FL, USA). For tumorigenicity characterization, the mice were monitored for survival. For therapeutic evaluations, the mice were randomly divided and received treatment of LHNPs (1 mg/mouse, intravenously) or TMZ (0.1 mg/mouse, intraperitoneally), or both starting from day 7 after tumor inoculation three times a week for 3 weeks. The mice were then monitored for survival daily, imaged using IVIS, and euthanized when neurological symptoms appeared.

**Statistical analysis**. In vitro experiments were performed with at least three independent biological samples unless stated otherwise. Data are presented as mean values ± standard deviations (SD). Differences between experimental groups were determined based on two-tailed student's $t$-test. Survival of animals in in vivo studies were analyzed based on Kaplan–Meier analyses and log-rank tests (WEHI Bioinformatics). A $P$-value < 0.05 were considered statistically significant.

**Reporting summary**. Further information on research design is available in the Nature Research Reporting Summary linked to this article.

## Data availability

The cDNA array and ChIP-seq data generated in this study have been deposited in the Gene Expression Omnibus (GEO) database under accession code GSE187418 and GSE196067, respectively. Remaining data are provided within the Article, Supplementary Information and Source Data. Source data are provided with this paper.

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

## Acknowledgements

J.L., X.W., A.T.C., and X.G contributed equally to this work. This work was supported by NIH Grant NS095817 (JZ) and CA245313 (RF, JZ).

## Author contributions

T.R.P., W.M.S and J.Z. conceived the project. J.L., X.W., A.T.C., X.G., B.T.H., H.Z., Z.C., J.W., W.C.S, and G.D. performed the experiments. J.L. and J.Z. wrote the paper, with the help of the co-authors. All authors read and approved the final manuscript.

## Competing interests

The authors declare no competing interests.
