## [Peer Review File · Nature Communications]

REVIEWER COMMENTS

Reviewer #1 (Remarks to the Author):

The study by Liu et al. uncovers and characterizes zinc finger protein 117 (ZNF117) as a regulator of glioblastoma stem cell differentiation. The authors found ZNF117 via a RNAi screen and showed that its knockdown/out leads differentiation towards the oligodendroglial lineage. They showed that ZNF117 inhibits Notch activation and binds to JAG2. They showed that inhibition of ZNF117 in vivo leads to inhibition of GBM xenograft growth. They concluded that ZNF117 is a differentiation regulator that acts via Notch and that can be targeted for therapy.

The findings of this study are new as ZNF117 has not been associated with GBM differentiation or any aspect of GBM malignancy before. The functional and experimental therapeutic data are convincing. The manuscript is well written. However, the following major issues are noted and should be addressed:

- 1- The entire study relies on knockdown/out. No overexpression experiments were performed. Considering the potential for non-specific effects for knockdown/out approaches, rescue experiments using ZNF117 exogenous expression should be performed to confirm the specificity of the knockdown/out findings.
- 2- The mechanistic experiments involving Notch are superficial and do not go far enough to show that Notch mediates the effects ZNF117 on differentiation. Notch expression/activation should be manipulated in the setting of ZNF117 knockdown/out/overexpression to demonstrate that Notch in fact mediates ZNF117-regulated differentiation.
- 3- How were the correlations of ZNF117 levels with patient survival analyzed? This is not explained in any detail. A quick TCGA analysis using cBioPortal by this reviewer showed no correlation with patient survival. This should be clarified.
- 4- The discussion is too short and uninformative. The implications and limitations of the work should be better discussed. What is the relevance of oligodendroglial vs astrocytic differentiation? The inhibition of ZNF117 for therapy in a clinical setting will be very difficult to achieve. These and other issues should be discussed.

Reviewer #2 (Remarks to the Author):

The manuscript is solid with important results relating to targets and treatments for GBM. Because many aspects, from biology to nanoparticle engineering to cancer models to therapeutic efficacy are presented, the article would be of interest to the wide and multidisciplinary readership of Nature Communications. I recommend accepting after revisions.

(1) Some data (e.g. Figure 1) is plotted using ID numbers designated as AB(AB00477)-xxx. It is unclear

from the text what this signifies. Are these the codes from the Dharmacon siGenome library? This could be further clarified in the text to help readers understand.

(2) Is there any concern that the pooled siRNAs each have varying levels of gene silencing ability? Have each of the sequences been verified for on-target activity?

(3) Do any of the sgRNAs targeting ZNF117 exhibit off target editing of other genes? Especially for sgZNF117-1 and sgZNF117-2, the two guide RNAs produce such different in vivo results, I suggest looking deeply into off target effects to ensure that only ZNF117 is being inhibited and not other genes that could add to the anti-tumor effects.

(4) The difference (Fig. 3g) between sgZNF117-1 and sgZNF117-2 is very dramatic and significant. All mice implanted with sgZNF117-1 treated cells are alive, while nearly all mice implanted with sgZNF117-2 treated cells die closer to sgGFP control mice. However, differences in proliferation (Fig. 3c), targeting, and other data is not different. The explanation given is that “sgZNF117-1 reduced the expression of ZNF117 at higher efficiency”, but I do not see data to support that conclusion. Fig. 3f shows some differences but not other readouts. I suggest investigating further.

(5) Please perform statistical analyses (comparisons) for the groups in Fig. 6e.

(6) Were unmodified or chemically modified sgRNAs used?

(7) Please add more details to the main text and/or figure captions about the nanoparticle delivery. How many injections, dosing schedule, what dose of sgRNA, injection route, etc. Readers will be interested in these details.

Reviewer #3 (Remarks to the Author):

In the present manuscript, the authors aimed at identifying novel therapeutic targets for glioblastoma (GBM). To this end, they performed an RNAi-based screening on GS5 cells and identified ZNF117 as candidate to promote GBM differentiation into oligodendrocyte lineage. GS5 cells with reduced expression of ZNF117 proliferated less and gave rise to less spheres with fewer cells, and showed an overall increased differentiation in oligodendrocytes as assessed by GalC staining. By RNAseq, they authors found Notch signaling to be regulated by ZNF117, and by ChIP-Seq revealed that ZNF117 regulated the promoter of JAG2, a Notch ligand, by preventing its activation, as shown via luciferase assay. Importantly, the authors showed an increased survival of mice either transplanted with GS5 cells engineered to express low levels of ZNF117, or following injection of nanoparticles, loaded with gRNA against ZNF117 together with CRISPR/Cas9, in nude mice transplanted with PS30 cells, sensitizing them for more conventional anticancer treatment.

The manuscript tackles a very important problem and is overall interesting, as it identified a candidate by in vitro cultures and showed its therapeutic potential in xenografts. However, there are several

points that need to be addressed:

_ In the section 2 of the results, the authors wanted to investigate the role of ZNF117 in differentiation. However, in this section, the authors analyzed by single-cell RNAseq on GS5 cells, without any manipulation of ZNF117. When looking at the differentially expressed genes, they found ZNF117 to be enriched along the oligodendroglial lineage. As such, the initial sentence of the first and third paragraph needs a substantial revision;

_ it is important to show the expression of ZNF117 in tSNE plot in Figure 2, along with the expression of other cell-type specific markers (e.g. Cspg4, Olig2 for oligo, Gfap, Sox9 for astros, Rbfox3, Snap25 etc.. for neurons, and not just 4 markers as in Supplementary Figure 2).

- Along this line, why do the authors use t-SNE and not UMAP?

_ Can the authors list more genes on the oligodendroglial lineage in figure 2f? Is it expressed in “early OPCs” (e.g. similar to Cspg4, CNPase) or also in late progenitors/oligodendrocytes (e.g. Mag, Mog etc...)? Which genes did they use to build the heatmap in figure 2e?

_ Page 6: what does “patient-specific” cell type markers mean? I guess “patient-specific” is wrong, as the authors selected “cell type specific markers”.

_ ZNF117 knock-out: can the authors add a panel in, e.g., supplementary figure 3, where they show their knock-down strategy? Where do the gRNAs bind? To promoter region, first exon?

- It's important that the authors show similar effects from 2 different gRNAs. However, they do not comment about possible off-targets. Can the authors show that the expression of putative off-targets is not affected, e.g. by qPCR?

- At page 10, the authors claim that “ZNF117 inhibits the proliferation”: as the proliferation seems rather “reduced”, it would be good to re-phrase the sentence;

_ Figure 3e: it would be important to show the quantification of the reduced expression of Nestin and increased expression of GalC in main figure, such as moving supplementary figure 3b. Can the authors show each biological replicate as dot? Can the authors repeat the experiment with another oligodendrocyte marker, such as NG2, or Sox10?

_ the authors treat the cells with FBS for 2-3 weeks before checking the expression of ZNF117. Can the authors describe the composition of the culture at the end of the experiment?

_ Figure 4a: the control lane (B-actin) seems different from ZNF117 lane. In particular, the samples 3 and 4 in control lane seem smaller than the corresponding ZNF117 lane. Can the authors provide the original blot? How many times was the WB repeated? It would be important to provide such information in the figure legend;

_ Figure 4d: same as before for figure 3e;

_ molecular characterization of ZNF117: the authors found differentially expressed genes between

control and ZNF117-KO cells. They authors focused on down-regulated genes, but half of differentially expressed genes is indeed upregulated. It would be very important to analyze these data in unbiased way (e.g. GO, GSEA) and discuss the upregulated genes, as it could further support the differentiation into oligodendroglial lineage, or identify putative novel mechanisms triggering GBM differentiation. Supplementary figure 6b seems too superficial for the dataset;

_ likewise, the authors performed ChIP-Seq on ZNF117, but they did not show any analysis; a comprehensive analysis (e.g. fraction of exons/introns/intergenic regions bound by ZNF117; sequence recognized by ZNF117 etc...) is required.

_ the authors found that the expression of Notch-related genes is reduced upon ZNF117 knock-out and that JAG2 promoter is regulated by ZNF117. Do the authors found JAG2 to be downregulated in their ZNF117-KO RNAseq? And, conversely, are the promoters of Notch-related genes (page.13) bound by ZNF117 in their ChIP-Seq? As for now, it seems that the authors imply that the downregulation of JAG2, determined by ZNF117, leads to the downregulation of the Notch pathway, but there is no data to support this conclusion. Ideally, the authors should investigate whether the knock-down of JAG2 is sufficient to downregulate the expression of Notch-related genes. At least, they should discuss this possibility, or cite papers where this has been investigated.

_ in Figure 5f it would be important to add the negative control of the CHIP; or does Y-axis depict fold change over control? If so, which control?

_ Nanoparticles to modulate ZNF117 in vivo: the strategy appears very interesting and promising. The authors checked the expression of ZNF117 in transplanted cells. However, for possible therapeutical approaches, it is essential to check the nanoparticles targets only tumor cells and not all the cells, or stem cells (e.g. in the SVZ or dentate gyrus) or oligodendrocyte progenitor cells.

Material and methods: they poorly described the experimental setup. Here some examples:

- in the Genome-wide RNAi, screen, did the authors test the knock-down of 18,119 separately? Does this mean they analyzed more than 300 plates?
- For the scRNA-seq: which protocol did they use? 10XGenomics? Drop-seq?
- Affymetrix-HuGene upon ZNF117 knockdown: after how many days in cultures were the cells collected? How was the RNA extracted? Samples prepared? How many biological replicates? How did the authors analyze the data? The authors refers to 2 previous publication, but it would be essential to mention here the procedure for the analysis.

In summary, the present manuscript is potentially interesting, but in the present form it lacks important controls (above all, the specificity of nanoparticle targeting to tumours) and many information in the material sections, and is not suited for publication.

** See Nature Research's author and referees' website at www.nature.com/authors for information about policies, services and author benefits.

Point-by-point response to Reviewers' Comments

We thank the reviewers for their thoughtful critiques, which we have used as the basis for an extensive revision of the manuscript. The Reviewer's comments are provided in italics. Our response to each comment is provided in plain text. Changes are marked in blue in the revised manuscript.

Reviewer #1:

The study by Liu et al. uncovers and characterizes zinc finger protein 117 (ZNF117) as a regulator of glioblastoma stem cell differentiation. The authors found ZNF117 via a RNAi screen and showed that its knockdown/out leads differentiation towards the oligodendroglial lineage. They showed that ZNF117 inhibits Notch activation and binds to JAG2. They showed that inhibition of ZNF117 in vivo leads to inhibition of GBM xenograft growth. They concluded that ZNF117 is a differentiation regulator that acts via Notch and that can be targeted for therapy. The findings of this study are new as ZNF117 has not been associated with GBM differentiation or any aspect of GBM malignancy before. The functional and experimental therapeutic data are convincing. The manuscript is well written. However, the following major issues are noted and should be addressed:

Response: We thank the reviewer for the positive comments on the novelty, experimental design, and presentation. We are grateful for all the constructive comments, which we have fully addressed as follows:

1- The entire study relies on knockdown/out. No overexpression experiments were performed. Considering the potential for non-specific effects for knockdown/out approaches, rescue experiments using ZNF117 exogenous expression should be performed to confirm the specificity of the knockdown/out findings.

Response: We appreciate the critical comment and have followed the advice to perform rescue experiments. We found ZNF117 exogenous expression reversed the phenotype changes, specifically oligodendroglial lineage differentiation, induced by CRISPR-mediated knockdown/out. The results are now presented in Supplementary Figure 5 and discussed on page 10 in the revised manuscript.

2- The mechanistic experiments involving Notch are superficial and do not go far enough to show that Notch mediates the effects ZNF117 on differentiation. Notch expression/activation should be manipulated in the setting of ZNF117 knockdown/out/overexpression to demonstrate that Notch in fact mediates ZNF117-regulated differentiation.

Response: Our data show that ZNF117 knockout promote GSC differentiation by regulating the expression of Notch ligand JAG2 through interaction with its promoter (Figure 5). Following the advice, we overexpressed JAG2 under a CMV promoter and found the overexpression reversed ZNF117 knockout-mediated oligodendroglial lineage differentiation. This finding, which is now presented in Supplementary Figure 5 and discussed on page 15 in the revised manuscript, provides further evidence that Notch mediates the differentiation effect of ZNF117.

3- *How were the correlations of ZNF117 levels with patient survival analyzed? This is not explained in any detail. A quick TCGA analysis using cBioPortal by this reviewer showed no correlation with patient survival. This should be clarified.*

Response: We apologize for the confusion. All the analyses were performed using Gliovis (<http://gliovis.bioinfo.cnio.es>). Detailed parameters for the analyses are as follows: For Figures a&b: Dataset: adult “TCGA_GBM”, Platform: “RNA-seq”, Cutoff value: “optimal cutoff”; For Figure c, Dataset: adult “Rembrandt”, Cutoff value: “optimal cutoff”; For Figure d, Dataset: adult “CGGA”, Tumor type: “Primary”, Histology: “All”; Cutoff value: “optimal cutoff”. We have included the information in the legend for Supplementary Figure 9 in the revised manuscript.

4- *The discussion is too short and uninformative. The implications and limitations of the work should be better discussed. What is the relevance of oligodendroglial vs astrocytic differentiation? The inhibition of ZNF117 for therapy in a clinical setting will be very difficult to achieve. These and other issues should be discussed.*

Response: We have followed this advice and included an additional paragraph of discussion in the revised manuscript on pages 20-21 in the revised manuscript.

Reviewer #2:

The manuscript is solid with important results relating to targets and treatments for GBM. Because many aspects, from biology to nanoparticle engineering to cancer models to therapeutic efficacy are presented, the article would be of interest to the wide and multidisciplinary readership of Nature Communications. I recommend accepting after revisions.

Response: Thanks for all the positive comments on the significance and experimental design. We are grateful for all the constructive comments, which we have fully addressed as follows:

(1) Some data (e.g. Figure 1) is plotted using ID numbers designated as AB(AB00477)-xxx. It is unclear from the text what this signifies. Are these the codes from the Dharmacon siGenome library? This could be further clarified in the text to help readers understand.

Response: We apologize for the confusion. AB(AB00477)-xxx represents the ID number of the specific siRNA coded by Dharmacon. This ID number is provided to allow others to order the specific sequence if needed. To avoid confusion, we have added clarification to the caption in Figure 1 in the revised manuscript.

(2) Is there any concern that the pooled siRNAs each have varying levels of gene silencing ability? Have each of the sequences been verified for on-target activity?

Response: The reviewer is correct there are variations among pooled siRNAs, which is shown in Supplemental Table 1. To address this concern, we tested each of 4 siRNAs separately in the validation study (Figure 1c&d). We did not verify on- or off- target activities of individual siRNA sequences, as we chose to use CRISPR in the follow-up experimental studies. The on-

target activities of selected guide sequences were characterized (Figure 3). Following your advice in Comment 3 below, we have determined the potential off-target effects. Please find our response to Comment 3 for details.

(3) Do any of the sgRNAs targeting ZNF117 exhibit off target editing of other genes? Especially for sgZNF117-1 and sgZNF117-2, the two guide RNAs produce such different in vivo results, I suggest looking deeply into off target effects to ensure that only ZNF117 is being inhibited and not other genes that could add to the anti-tumor effects.

Response: We appreciate the critical suggestion. Following your advice, we carefully analyzed sgZNF117-1, which demonstrated significantly greater efficacy than sgZNF117-2 in vivo (as discussed in Comment 4 below). We found there are no sequences having 0 or 1 mismatches with sgZNF117-1 throughout the genome and 179 sequences having 2-4 mismatches. We sequenced all the 6 genes carrying mismatched sequences in exons and found that, other than 1 that is not expressed in this specific cell line, the other 5 have no detectable disruption. These findings suggest that the observed anti-tumor effects of sgZNF117-1 is unlikely to be caused by off-target effects. We have included the analysis and findings in Supplementary Figure 6 and discussed on page 10 in the revised manuscript.

(4) The difference (Fig. 3g) between sgZNF117-1 and sgZNF117-2 is very dramatic and significant. All mice implanted with sgZNF117-1 treated cells are alive, while nearly all mice implanted with sgZNF117-2 treated cells die closer to sgGFP control mice. However, differences in proliferation (Fig. 3c), targeting, and other data is not different. The explanation given is that “sgZNF117-1 reduced the expression of ZNF117 at higher efficiency”, but I do not see data to support that conclusion. Fig. 3f shows some differences but not other readouts. I suggest investigating further.

Response: As described in the response to the above comment, we performed sequencing analysis and did not identify significant off-target effects from sgZNF117-1. We hope that the reviewer can appreciate the significant differences between the two sgRNA sequences in down-regulating ZNF117 expression (Figure 3b) as well as in promoting oligodendroglial lineage differentiation (Figure 3e-g). We agree that the difference in proliferation (Figure 3c) appears to be small, likely because that short term cell proliferation is not the best measurement for differentiation. To avoid confusion to readers, we have analyzed potential off-target effects of sgZNF117-1 and added our findings and discussion on page 10 in the revised manuscript.

(5) Please perform statistical analyses (comparisons) for the groups in Fig. 6e.

Response: We have followed your advice and performed statistical analyses (comparisons) for the groups in Figure 6e. Specific findings include: sgGFP vs. sgZNF117, $P=0.007$; sgZNF117 vs. sgZNF117+TMZ, $P=0.015$; TMZ vs sgZNF117+TMZ, $P=0.004$. The information is now included on page 19 in the revised manuscript.

(6) Were unmodified or chemically modified sgRNAs used?

Response: Modified sgRNAs synthesized by Synthego were used in this study. The information is now included on page 25 in the revised manuscript.

(7) Please add more details to the main text and/or figure captions about the nanoparticle delivery. How many injections, dosing schedule, what dose of sgRNA, injection route, etc. Readers will be interested in these details.

Response: We have followed the advice and included all the relevant details on page 17 in the revised manuscript.

Reviewer #3:

In the present manuscript, the authors aimed at identifying novel therapeutic targets for glioblastoma (GBM). To this end, they performed an RNAi-based screening on GS5 cells and identified ZNF117 as candidate to promote GBM differentiation into oligodendrocyte lineage. GS5 cells with reduced expression of ZNF117 proliferated less and gave rise to less spheres with fewer cells, and showed an overall increased differentiation in oligodendrocytes as assessed by GalC staining. By RNAseq, they authors found Notch signaling to be regulated by ZNF117, and by ChIP-Seq revealed that ZNF117 regulated the promoter of JAG2, a Notch ligand, by preventing its activation, as shown via luciferase assay. Importantly, the authors showed an increased survival of mice either transplanted with GS5 cells engineered to express low levels of ZNF117, or following injection of nanoparticles, loaded with gRNA against ZNF117 together with CRISPR/Cas9, in nude mice transplanted with PS30 cells, sensitizing them for more conventional anticancer treatment.

The manuscript tackles a very important problem and is overall interesting, as it identified a candidate by in vitro cultures and showed its therapeutic potential in xenografts. However, there are several points that need to be addressed:

Response: We thank the reviewer for the positive comments on significance of the work. We are grateful for all the constructive comments, which we have fully addressed as follows:

1. - In the section 2 of the results, the authors wanted to investigate the role of ZNF117 in differentiation. However, in this section, the authors analyzed by single-cell RNAseq on GS5 cells, without any manipulation of ZNF117. When looking at the differentially expressed genes, they found ZNF117 to be enriched along the oligodendroglial lineage. As such, the initial sentence of the first and third paragraph needs a substantial revision;

Response: We appreciate the constructive comments and have revised the writing accordingly. Specifically, we have changed the following sentences. For the first paragraph: Original writing: To further identify which cell lineage is regulated by ZNF117-mediated GSC differentiation, we performed scRNA-seq on GS5 cells. Revised writing: To determine which genes regulate GSC differentiation into terminal cell types, we performed scRNA-seq on GS5 cells.

For the third paragraph: Original writing: Next, we performed single-cell trajectory analysis to determine into which lineage ZNF117 mediates GSC differentiation. Revised writing: Next, we

constructed single-cell trajectories from GSCs and differentiated cells to determine which genes mediate GSC differentiation into specific lineages

2. - *it is important to show the expression of ZNF117 in tSNE plot in Figure 2, along with the expression of other cell-type specific markers (e.g. Cspg4, Olig2 for oligo, Gfap, Sox9 for astros, Rbfox3, Snap25 etc.. for neurons, and not just 4 markers as in Supplementary Figure 2).*

Response: Thanks for the suggestion. In the revised manuscript, we have followed your advice and revised the figure, which is now presented in UMAP instead of tSNE based on your Comment 3. The expression of ZNF117 and other cell-type specific markers (NES-GSC, GFAP-astrocytes, PLP1-oligodendrocytes, SNAP25-neurons) is shown on Figure 2b.

3. - *Along this line, why do the authors use t-SNE and not UMAP?*

Response: t-SNE was used because at that time when the single-cell trajectory analysis was performed, UMAP was not yet available. In the revised manuscript, we included updated Figure 2a, which shows GS5 cells visualized with UMAP.

4. - *Can the authors list more genes on the oligodendroglial lineage in figure 2f? Is it expressed in “early OPCs” (e.g. similar to Cspg4, CNPase) or also in late progenitors/oligodendrocytes (e.g. Mag, Mog etc...)? Which genes did they use to build the heatmap in figure 2e?*

Response: We appreciate all the suggestions. In the revised manuscript, we have included additional OPC genes (CSPG4, CNP), an oligodendrocyte gene (PLP1), and GSC genes (KLF4, MYC, SOX2) to the heatmap in updated Figure 2f. Genes that significantly regulate GSC to oligodendrocyte differentiation were used to build the heatmap in Figure 2e (now 2f). The information is now included on page 8 to increase clarity.

5. - *Page 6: what does “patient-specific” cell type markers mean? I guess “patient-specific” is wrong, as the authors selected “cell type specific markers”.*

Response: We apologize for the confusion. “Patient-specific” refers an individual gene marker used to analyze a cell type in an individual sample. To avoid confusion, we have modified the writing as follows: To identify specific cell populations, we used cell type markers that had been reported in previous studies to be related to specific cell types. The caption for Extended Data Figure 2b has been modified to increase clarity.

6. - *ZNF117 knock-out: can the authors add a panel in, e.g., supplementary figure 3, where they show their knock-down strategy? Where do the gRNAs bind? To promoter region, first exon?*

Response: We have followed the advice and included a panel to show the knock-down strategy in Supplementary Figure 4a in the revised manuscript.

7. - *It’s important that the authors show similar effects from 2 different gRNAs. However, they do not comment about possible off-targets. Can the authors show that the expression of putative off-targets is not affected, e.g. by qPCR?*

Response: Thanks for the critical comment. Following your suggestion, we carefully examined sgZNF117-1, which demonstrated greater biological effects than sgZNF117-2. We found there are no sequences having 0 or 1 mismatches with sgZNF117-1 throughout the genome and 179 sequences having 2-4 mismatches. We sequenced all the 6 genes carrying mismatched sequences in exons and found that, other than 1 that is not expressed in this specific cell line, the other 5 have no detectable disruption. These findings suggest there are no significant off-target effects. This analysis is now presented in Supplementary Figure 6 and discussed on page 10 in the revised manuscript.

8. - *At page 10, the authors claim that “ZNF117 inhibits the proliferation”: as the proliferation seems rather “reduced”, it would be good to re-phrase the sentence;*

Response: We appreciate the comment and revised the sentence accordingly.

9. *_ Figure 3e: it would be important to show the quantification of the reduced expression of Nestin and increased expression of GalC in main figure, such as moving supplementary figure 3b. Can the authors show each biological replicate as dot? Can the authors repeat the experiment with another oligodendrocyte marker, such as NG2, or Sox10?*

Response: We agree with the reviewer and have moved the quantification figure to Figure 3 as 3f, in which each biological replicate is shown as a dot. We have also followed the advice and repeated the experiment with a second oligodendrocyte marker, Olig1. The finding, which is now included in Supplementary Figure 4c and discussed on page 10, is consistent with the finding based on GalC analysis.

10. *_ the authors treat the cells with FBS for 2-3 weeks before checking the expression of ZNF117. Can the authors describe the composition of the culture at the end of the experiment?*

Response: We have followed the advice and analyzed GS5 cells after culturing in FBS-containing medium. The results, which are now included in Supplementary Figure 3 and discussed on page 9 in the revised manuscript, suggest that treatment with FBS fully differentiated GS5 cells and significantly reduced Nestin+ GSCs.

11. *_ Figure 4a: the control lane (B-actin) seems different from ZNF117 lane. In particular, the samples 3 and 4 in control lane seem smaller than the corresponding ZNF117 lane. Can the authors provide the original blot? How many times was the WB repeated? It would be important to provide such information in the figure legend;*

Response: Thanks for the comment. The original blot image is now included in Source Data. The WB was repeated two times. We have included this information on pages 12 (text) and 13 (legend) in the revised manuscript.

12. *_ Figure 4d: same as before for figure 3e;*

Response: We have followed the advice and revised the figures accordingly. Figure 4e now shows quantification with each replicate as a dot.

13. *_ molecular characterization of ZNF117: the authors found differentially expressed genes between control and ZNF117-KO cells. They authors focused on down-regulated genes, but half of differentially expressed genes is indeed upregulated. It would be very important to analyze these data in unbiased way (e.g. GO, GSEA) and discuss the upregulated genes, as it could further support the differentiation into oligodendroglial lineage, or identify putative novel mechanisms triggering GBM differentiation. Supplementary figure 6b seems too superficial for the dataset;*

Response: We apologize for the confusion. The major purpose of the cDNA array study is to identify signaling pathways altered by ZNF117 downregulation. Results presented in Figure 5a and Supplementary Figure 6b (now Supplementary Figure 8a in the revised manuscript) were supported by unbiased analyses of both these genes up-regulated and down-regulated. Based on the findings presented in Supplementary Figure 8a,b, we chose to focus on the Notch signaling. We agree it is confusing to mark only those Notch-related genes in the down-regulated cohort in Figure 5b. In the revised manuscript, we have included those both down- and up-regulated. We agree it would be interesting to have more analysis on the cDNA array dataset. As we hope not to distract our focus on the Notch pathway in this specific study, we would prefer not to include additional analyses.

14. *_ likewise, the authors performed ChIP-Seq on ZNF117, but they did not show any analysis; a comprehensive analysis (e.g. fraction of exons/introns/intergenic regions bound by ZNF117; sequence recognized by ZNF117 etc...) is required.*

Response: We appreciate the comment and have included a pie chart with all the suggested information in Supplementary Figure 8d.

15. *_ the authors found that the expression of Notch-related genes is reduced upon ZNF117 knock-out and that JAG2 promoter is regulated by ZNF117. Do the authors found JAG2 to be downregulated in their ZNF117-KO RNAseq? And, conversely, are the promoters of Notch-related genes (page.13) bound by ZNF117 in their ChIP-Seq? As for now, it seems that the authors imply that the downregulation of JAG2, determined by ZNF117, leads to the downregulation of the Notch pathway, but there is no data to support this conclusion. Ideally, the authors should investigate whether the knock-down of JAG2 is sufficient to downregulate the expression of Notch-related genes. At least, they should discuss this possibility, or cite papers where this has been investigated.*

Response: We appreciate the constructive comments. Yes, we did find that JAG2 is downregulated in ZNF117-KO cells (Figure 5g) and its promoter is directly bound by ZNF117 in ChIP-Seq analysis (Figure 5e, h, i). It was documented in literature that downregulation of JAG2 significantly reduces Notch activation through regulation of Notch-relevant genes, including Notch 1 (Casey LM. et al. Developmental Dynamics, 2006; Asnaghi L. et al. Oncotarget, 2016). To further provide evidence to support our observation that ZNF117 regulates Notch signaling

through JAG2, we overexpressed JAG2 under promoter CMV and found the overexpression reversed ZNF117 knockout-mediated oligodendroglial lineage differentiation. In the revised manuscript, we included these finding in Supplementary Figure 5 and relevant discussion on pages 14-15 with the citations.

16. *_ in Figure 5f it would be important to add the negative control of the CHIP; or does Y-axis depict fold change over control? If so, which control?*

Response: We apologize for the confusion. The label the Y-axis of Figure 5f was mislabeled and should be the ratio of CHIP signal in the negative control receiving treatment of sgGFP vs. ZNF117-knockout group. The figure is now corrected in the revised manuscript.

17. *_ Nanoparticles to modulate ZNF117 in vivo: the strategy appears very interesting and promising. The authors checked the expression of ZNF117 in transplanted cells. However, for possible therapeutical approaches, it is essential to check the nanoparticles targets only tumor cells and not all the cells, or stem cells (e.g. in the SVZ or dentate gyrus) or oligodendrocyte progenitor cells.*

Response: We appreciate the comment. The nanoparticle approach has been extensively characterized in our previous studies (Han L. et al. ACS Nano, 2016; Zhou Y. et al. Advanced Science, 2020; Chen Y. et al. Nature Cell Biology, 2020), in which we showed that nanoparticles after intravenous administration preferentially transfected tumor cells but not others. This is due to two major reasons. First, the nanoparticles are designed to target tumors through the interaction of iRGD with tumor vasculature. Second, compared to normal brain cells, tumor cells are proliferative and easy to be transfected. In the revised manuscript, we further characterized the nanoparticles in the PS30 tumor xenografts. We found that, consistent with our previous findings, the nanoparticles selectively transfected tumor cells but not the normal brain. The results are now presented in Supplementary Figure 10 and discussed on page 17 in the revised manuscript.

18. *Material and methods: they poorly described the experimental setup. Here some examples: - in the Genome-wide RNAi, screen, did the authors test the knock-down of 18,119 separately? Does this mean they analyzed more than 300 plates? - For the scRNA-seq: which protocol did they use? 10XGenomics? Drop-seq? - Affymetrix-HuGene upon ZNF117 knockdown: after how many days in cultures were the cells collected? How was the RNA extracted? Samples prepared? How many biological replicates? How did the authors analyze the data? The authors refers to 2 previous publication, but it would be essential to mention here the procedure for the analysis.*

Response: Yes, we tested 18,119 genes separately and in total we analyzed more than 300 plates. The scRNA-seq was performed using a microchip-based platform called scFTD-seq, which was developed by the Fan group and described in this study (Dura B. et al. Nucleic Acids Research, 2019). We have followed the suggestions and included all the details in the relevant sections on pages 23 and 26.

REVIEWERS' COMMENTS

Reviewer #1 (Remarks to the Author):

The authors satisfactorily addressed this reviewer's critique.

Reviewer #2 (Remarks to the Author):

All of my comments were adequately addressed. I support this manuscript for publication (accept).

Reviewer #3 (Remarks to the Author):

The present manuscript is a much improved version of the initially submitted manuscript. The authors addressed most of my points convincingly. A deeper analysis of RNA-seq and ChIP-seq would have added further value to the manuscript, but it's understandable that this was not included due to space constrain and to keep the focus of the manuscript.

A small detail: at pag.15 figure showing the down regulation of JAG2 upon ZNF117 is Fig.5g and not Fig.5f, as indicated in the text.

Overall, the subject is relevant, the experiments well performed and the conclusions are supported by the data presented throughout the manuscript.

** See Nature Research's author and referees' website at www.nature.com/authors for information about policies, services and author benefits

Reviewer #3:

The present manuscript is a much improved version of the initially submitted manuscript. The authors addressed most of my points convincingly. A deeper analysis of RNA-seq and ChIP-seq would have added further value to the manuscript, but it's understandable that this was not included due to space constrain and to keep the focus of the manuscript.

A small detail: at pag.15 figure showing the down regulation of JAG2 upon ZNF117 is Fig.5g and not Fig.5f, as indicated in the text.

Overall, the subject is relevant, the experiments well performed and the conclusions are supported by the data presented throughout the manuscript.

Response: We are grateful for the reviewer's the positive comments and thank the reviewer for pointing out the typo, which has now been corrected in the revised manuscript.